## THE NATURAL HISTORY OF MODEL ORGANISMS

# The rhesus macaque as a success story of the Anthropocene

**Abstract** Of all the non-human primate species studied by researchers, the rhesus macaque (*Macaca mulatta*) is likely the most widely used across biological disciplines. Rhesus macaques have thrived during the Anthropocene and now have the largest natural range of any non-human primate. They are highly social, exhibit marked genetic diversity, and display remarkable niche flexibility (which allows them to live in a range of habitats and survive on a variety of diets). These characteristics mean that rhesus macaques are well-suited for understanding the links between sociality, health and fitness, and also for investigating intra-specific variation, adaptation and other topics in evolutionary ecology.

**EVE B COOPER\*, LAUREN JN BRENT, NOAH SNYDER-MACKLER, MEWA SINGH, ASMITA SENGUPTA, SUNIL KHATIWADA, SUCHINDA MALAIVIJITNOND, ZHOU QI HAI AND JAMES P HIGHAM**

**\*For correspondence:**
eve.cooper@nyu.edu

**Competing interest:** The authors declare that no competing interests exist.

## Introduction

Growing acceptance of the theory of evolution in the late 1800s led to an increased interest in studying non-human primates in order to better understand our own biology. It was during this period that the first scientific studies of the rhesus macaque (*Macaca mulatta*) were performed. The first paper described the "anatomy of advanced pregnancy" in rhesus macaques and was published in 1893 (*Hart and Gulland, 1893*), and a paper on behavioral cognitive experiments followed in 1902 (*Kinnaman, 1902*). However, the rhesus macaque really came to the forefront of non-human primate research in 1925 when the Carnegie Science Institute set up a breeding population to study embryology and fertility in a species similar to humans (*Wilson, 2012*). Rhesus macaques were found to be relatively easy to keep and breed in captivity and, at the time, they were also readily available via export from colonial India. All this led to increased interest in using rhesus macaques in research.

In 1938, the American primatologist Clarence Ray Carpenter released 409 Indian-origin rhesus macaques onto Cayo Santiago, a small island off the coast of Puerto Rico, for behavioral research (*Rawlins and Kessler, 1986*). This free ranging population has grown to over a thousand individuals, and has produced a wealth of research on rhesus macaque psychology and behavior that is likely unparalleled to our knowledge on any other primate species, with the exception of humans (*Box 1*).

Today, rhesus macaques make up 65% of the non-human primate research subjects in the United States (*Feister, 2018*), and they are likely the most studied non-human primate globally. Within their natural range, many populations of rhesus macaques also seem to be thriving. Their success appears to be occurring despite, or perhaps even owing to, the Anthropocene epoch: over the past 100 or so years rhesus macaque habitat has been characterized by rapid agricultural and urban shifts in landscape due to human intervention. Here we will focus on the considerable amount of basic research that has been done on rhesus macaques across fields of evolutionary biology, ecology, psychology and physiology, with special focus on how this research has gleaned insight into the natural history of the species.

## Evolutionary history

Rhesus macaques are Asian monkeys, sharing a common ancestor with humans roughly

## Box 1. The Cayo Santiago rhesus macaques.

The most intensively studied population of rhesus macaques is a free-ranging population of Indian origin that inhabit the 15.2 hectare tropical island of Cayo Santiago, one kilometer off the southeast coast of Puerto Rico (18° 09' N 65° 44' W). All rhesus macaques on the island are the descendants of 409 animals collected over a 12-district area (comprising 2500 km²) in the mountains near Lucknow, India in 1938 (*Rawlins and Kessler, 1986*). The monkeys were originally brought to the island by primatologist Clarence Ray Carpenter for behavioral research.

The population dipped in the 1950s causing a bottleneck whereby all individuals alive today are the descendants of 15 females that were alive in 1956 (*McMillan and Duggleby, 1981*). Today the population size is maintained at approximately 1500 individuals. Unsurprisingly, given the relatively small founder population and subsequent population bottleneck, the Cayo Santiago population is less genetically diverse then wild Indian rhesus macaques (*Kanthaswamy et al., 2017*). However, despite the reduced genetic diversity, there is no evidence of inbreeding depression in the population, with individuals seemingly outbred by disassortative mating, through which individuals with different phenotypes reproduce more frequently than would be expected under random mating (*Widdig et al., 2017*). In addition to eating both natural and introduced flora on the island, the monkeys are provisioned with commercial monkey chow (0.23 kg/animal/day) and water is available ad libitum. Aside from food provisioning, there are minimal interventions into the monkeys' lives. They are left to form their own social groups and mating pairs, and virtually no veterinary intervention is provided. All animals are trapped as juveniles and given ear notches and tattoos which allow for individual identification throughout their lives. Since 1956, a daily census has been conducted such that births, deaths, and changes in social group membership are known within approximately a two-day accuracy. Systematic collection of complete skeletal remains began in 1971, which, coupled with demographic and life-history data available on these individuals, is a valuable resource for a wide variety of research avenues (*Rawlins and Kessler, 1986*). Since 1992, genetic testing has also been conducted through a variety of methods, with thousands of individuals now genotyped (reviewed in *Widdig et al., 2016*), and the full pedigree has been resolved for the past three generations of monkeys.

The depth and breadth of data collected on the population, coupled with the ease with which they are observable in a free-ranging capacity, has led to their intensive study across a variety of disciplines from behavior, physiology, demography, ecology, genomics, and psychology. Consequently, the majority of the literature available on rhesus macaque behavior and life-history comes from this island population. The differences between the Cayo Santiago population and those occurring within the species' natural range in Asia have been reviewed elsewhere (*Maestripieri and Hoffman, 2012*), but generally they are found to be comparable in most aspects of social organization, behavior, and life-history.

The Cayo Santiago population provided an exemplar of rhesus macaques' characteristic resilience during a recent natural experiment where category 4 Hurricane Maria touched down on Cayo Santiago. Following this extreme weather event, individuals increased their number of social connections on the island, illustrating how the species may use social buffering to mitigate the negative consequences of major anthropogenic events (*Testard et al., 2021*). Additionally, while hurricanes on the island population have had a significant effect on patterns of gene expression (*Watowich et al., 2022*) , female rhesus macaques have been shown to increase their fertility later in life such that their lifetime reproductive success is not affected by these events (*Morcillo et al., 2020*).

25 million years ago (Ma; *Perelman et al., 2011*). The *Macaca* genus is a large, geologically recent radiation containing over 20 different species (*Fooden, 1976*; *Roos et al., 2019*, *Figure 1*).

The phylogenetic history of the macaque genus has been notoriously difficult to ascertain given the rapid diversification and significant introgression between macaque species (*Liedigk*

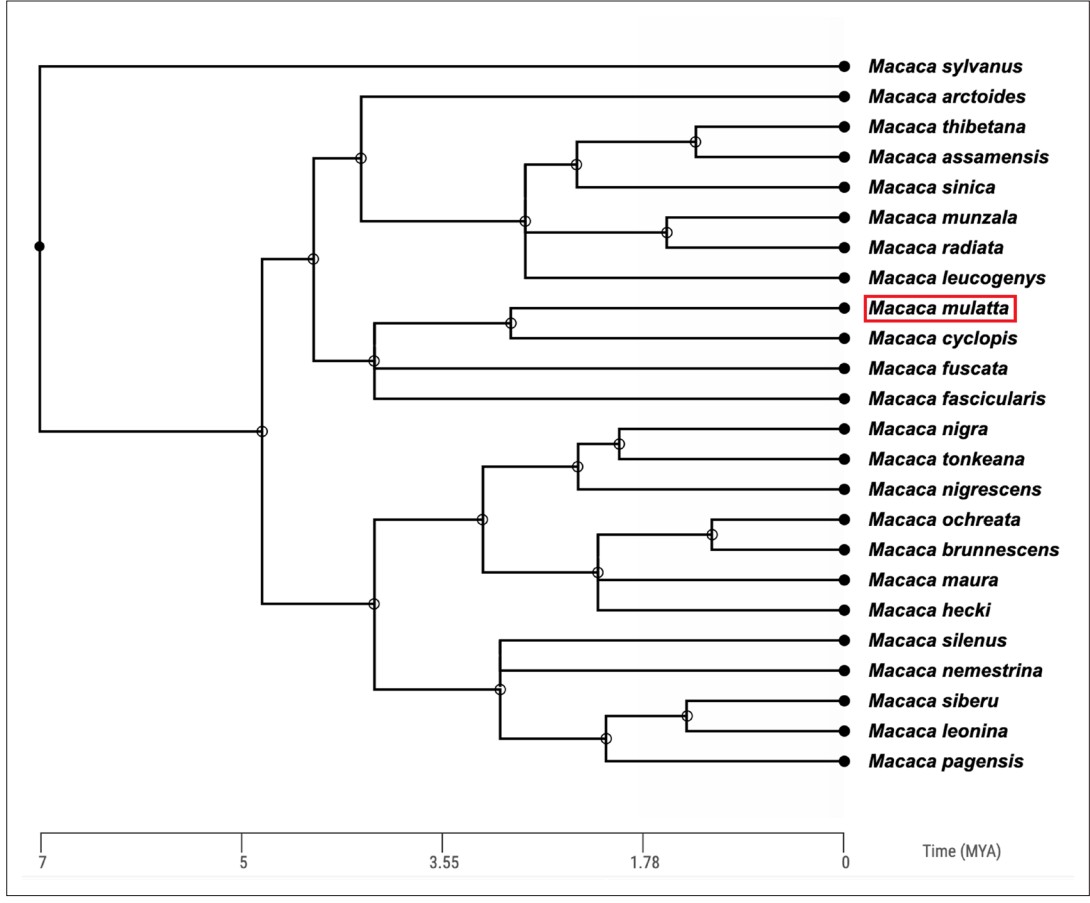

**Figure 1.** Time-calibrated phylogenetic tree of the 24 macaque (*Macaca*) species based on molecular data. The *Macaca* genus diverged from other members of the primate tribe *Papionini* approximately 7 million years ago. The position of the rhesus macaque (*Macaca mulatta*) is highlighted by the red box. Phylogeny constructed using TimeTree (**Kumar et al., 2017**).

*et al., 2014*; *Fan et al., 2018*; *Roos et al., 2019*; *Osada et al., 2021*). For example, based on mitochondrial DNA sequences, Chinese rhesus macaques appear more similar to Taiwanese macaques (*Macaca cyclopis*) and Japanese macaques (*Macaca fuscata*) than to Indian rhesus macaques, contradicting the taxonomic classification of the rhesus macaque species (*Roos et al., 2019*). However, based on the most complete nuclear genomic data currently available, a phylogeny of rhesus macaques and six of their most closely related sister species indicates that Indian and Chinese rhesus macaques are most closely related to one another, followed by Taiwanese macaques (*Macaca cyclopis*) and Japanese macaques (*Macaca fuscata*; *Figure 1*, *Osada et al., 2021*).

The discordance between phylogenies created using nuclear genomes and mitochondrial genomes is most likely a consequence of the significant ancient admixture between Chinese rhesus macaques and Taiwanese macaques as

well as Japanese macaques (*Roos et al., 2019*; *Osada al., 2021*). This admixture likely occurred during glacial episodes which created land bridges between China and Japan, and China and Taiwan, during the Pleistocene and early Holocene (*Marmi et al., 2004*).

In addition to the Taiwanese and Japanese macaques, there is evidence for admixture of the rhesus macaque genome and many other extant macaque species, including the stump-tailed macaque (*Fan et al., 2018*), the Tibetan macaque (*Macaca thibetana*; *Fan et al., 2014*), and the long-tailed macaque (*Higashino al., 2012*; *Yan et al., 2011*; *Bunlungsup et al., 2017*; *Ito et al., 2020*; *Satkoski Trask et al., 2013*). The high levels of relatively recent admixture between rhesus macaques and other macaque species makes them an excellent system through which we might gain a better understanding of ancient hybridization between different hominin species (*Boel et al., 2019*; *Buck et al., 2021*).

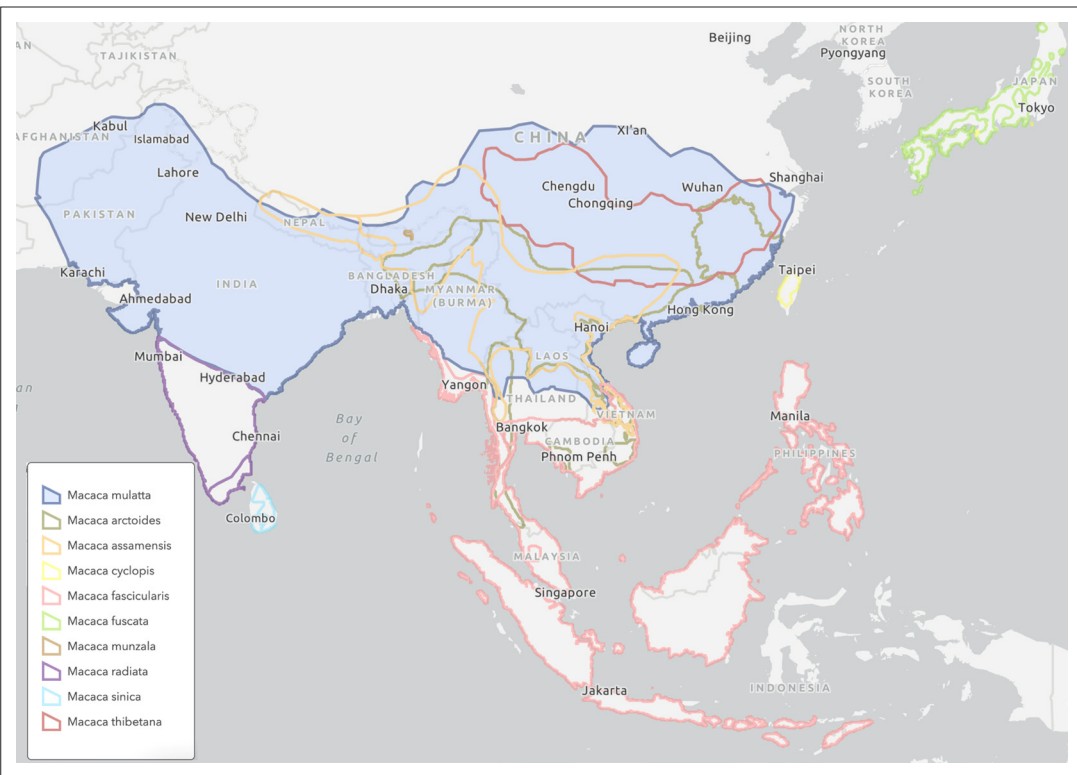

**Figure 2.** Geographical range across South and South-East Asia of rhesus macaques (*Macaca mulatta*) and 9 other extant sister species in the *Macaca* genus. The 10 species shown in different colors represent a complete monophyletic group, with the exception of *Macaca leucogenys*, which is missing because its complete range information is not yet known with high confidence. Rhesus macaques (blue shaded area) have the largest natural range of any non-human primate, which stretches from Afghanistan in the west, through Pakistan, India, Nepal, Bangladesh, Bhutan, Myanmar, Thailand, Laos, Vietnam, and across a large swathe of China in the east. Figure based on ICUN Red List species range estimates.

Rhesus macaques, a relatively new species with many closely-related competitors, have come to successfully populate a vast range across more than 6000 km and 11 countries within mainland Asia (*Figure 2*). The effective population size of rhesus macaques across their entire modern-day range is not known, but current effective population sizes of Chinese and Indian rhesus macaques are estimated at 240,000 and 17,000, respectively (*Hernandez et al., 2007*). These countries represent only a portion of total rhesus macaques, but even those population sizes alone indicate that rhesus macaques have a substantially larger population than other closely related macaque species in Asia (*Marmi et al., 2004*; *Bonhomme et al., 2009*). The large population of rhesus macaques is undoubtedly related to the extensive geographic area they occupy, which is the second largest geographic range of any primate species except humans.

## Intra-specific diversity and variation

Taxonomists have long recognized that rhesus macaques can be subdivided into distinct regional subpopulations (*Fooden, 2000*; *Gibbs et al., 2007*; *Pocock, 1932*; *Smith and McDonough, 2005*). While it is generally agreed that these populations are not sufficiently distinct to warrant a multi-species designation (*Fooden, 2000 Pocock, 1932*), the variation between two populations in particular has become the subject of intense research interest in the biomedical field. India, originally the primary source of rhesus macaques for research, suspended exportation in 1978, and research facilities began importing rhesus from China. Researchers then quickly began to recognize the need to understand how differences between the Indian-origin and Chinese-origin rhesus macaques might influence their studies. Wright's Fst value, a measure of the amount of genetic variance that can be explained

by population structure, is 0.14 between Indian- and Chinese-origin rhesus macaques. Relatedly, low levels of apparent admixture suggest that recurrent migration between the two groups has been minimal over the past 1.6 Ma (*Hernandez et al., 2007*).

Comparative research on these rhesus macaque populations includes attempts to understand how functional genetic variation impacts physiology, immunology, and behavior. There are differences between Chinese-origin and Indian-origin rhesus in disease pathogenesis, blood chemistry, the major histocompatibility complex, and aspects of behavior and temperament (*Champoux et al., 1994*; *Joag et al., 1994*; *Champoux et al., 1996*; *Binhua et al., 2002*; *Liu et al., 2008*; *Ma et al., 2009*; *Solomon et al., 2010*; *Jiang et al., 2013*). These differences have mainly been characterized using captive animals, and are not well understood in the context of ecological variation. Additionally, populations outside of China and India remain understudied by comparison (but see *Smith and McDonough, 2005*; *Hasan et al., 2014*; *Kyes et al., 2006*; *Ito et al., 2020*).

While the goal of this comparative research between different populations was predominately to understand variation in the translational value of Chinese-origin, Indian-origin, and hybrid rhesus macaques, it provides a thorough empirical basis for understanding the links between functional genomic variation and variation in ecology. Rhesus macaques have levels of single nucleotide polymorphisms (SNPs) that are exceptionally high among model organisms and approximately twice as high as those observed in most human populations (*Warren et al., 2020*). This extent of genetic diversity can create issues for the broad applicability of study results in the biomedical sciences given the lack of isogenic lines in the species, but is a boon to those interested in mapping genotype to phenotype relationships (*Champoux et al., 1996*; *Smith and McDonough, 2005*; *Liu et al., 2008*; *Ma et al., 2009*; *Xue et al., 2016*).

Functional genetic studies on rhesus macaques have been used to identify specific genetic mechanisms underlying a wide variety of physiological and behavioral traits, including the timing of male natal dispersal (*Trefilov et al., 2000*), the degree of female multimale mating (*Trefilov et al., 2005*), the oxytocin response and maternal behavior in lactating mothers (*Wood et al., 2022*), and adaptations to cold climatic conditions at the northern edge of the species range (*Liu et al., 2018*).

## Ecology and diet

Rhesus macaques persist over a remarkable range of environments: their niche includes both tropical and temperate climates, elevations from sea level to over 4000 meters, and forested, semi-desert, and swamp habitats (*Neville, 1968*; *Lindburg, 1971*; *Goldstein and Richard, 1989*; *Richard et al., 1989*; *Liu et al., 2018*). This flexibility in niche is highly unusual among non-human primate species, which often occupy narrow niche ranges. Although they are able to thrive under a variety of conditions, rhesus macaques are found at the highest densities in non-forested habitat (*Fooden, 2000*), suggesting that their most preferred habitat is an open environment with sparse tree cover (*Richard et al., 1989*). This is likely a consequence of having evolved alongside frequent Pleistocene glacial episodes, which resulted in rhesus macaques developing their niche within 'disturbed environments' - those with only partial or secondary forest growth, and, more recently, in urbanized environments (*Saraswat et al., 2015*).

Rhesus macaques attain adult lengths of 40–60 cm, and weights of 4–10 kg (*Fooden, 2000*). Among macaque species they have a low to medium degree of sexual dimorphism in body size, with adult males being on average 51% larger than adult females (*Turcotte et al., 2021*). In accordance with Bergmann's rule, which states that organisms at higher latitude should be larger in order to conserve heat, within the rhesus macaque species body size does increase with latitude across their natural range, with average male body size increasing 100% and average female body size increasing 75% from the southernmost point (15 °N) to the northernmost point (35 °N) of their range (*Fooden, 2000*). One exception to Bergmann's rule occurs among rhesus macaque populations in the Indochinese hybrid zone, which has seen recent introgression with long-tailed macaques (see Evolutionary History), resulting in smaller body sizes relative to their latitude (*Bunlungsup et al., 2017*).

Rhesus macaques are generalist omnivores, and have a highly varied and flexible diet (*Figure 3*). They are primarily vegetarian and their diets typically reflect whatever food sources are predominant in their environment (*Lindburg, 1971*; *Tang et al., 2016*; *Zhou et al., 2014*). In tropical and subtropical forests, rhesus macaques have been found to be primarily frugivorous, while in temperate and limestone forests they have been described as generally folivorous (*Lindburg, 1977*; *Richard et al., 1989*; *Sengupta and*

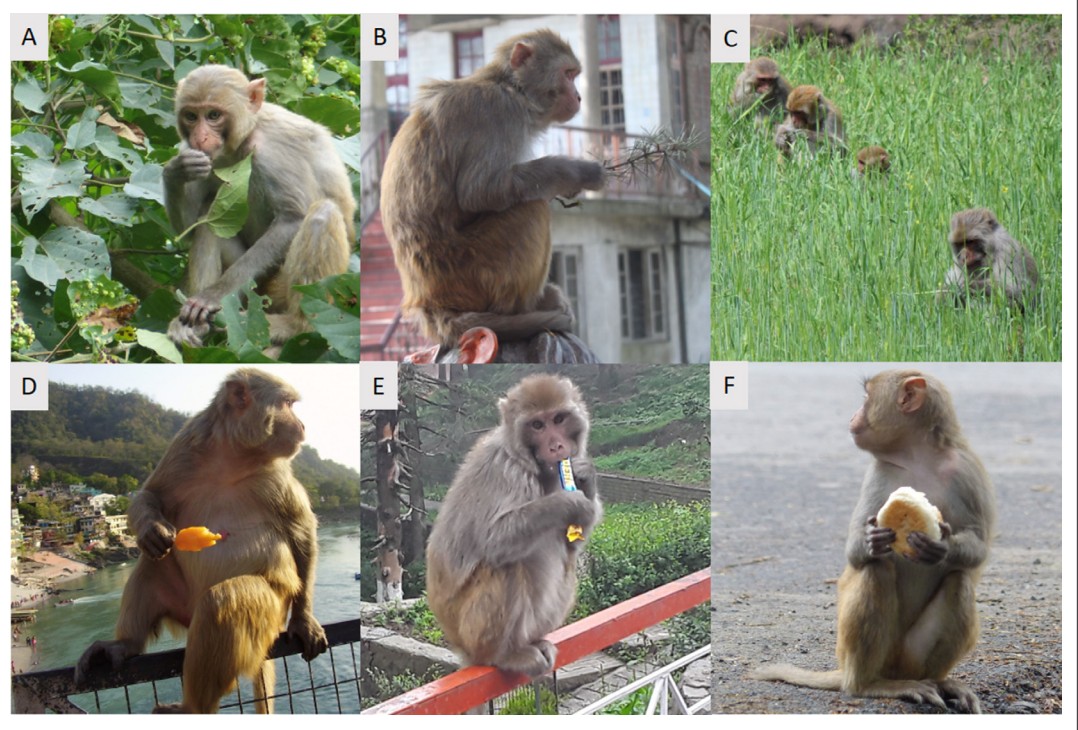

**Figure 3.** The diets of rhesus macaques can include both natural and anthropogenic food sources. Rhesus macaques are shown eating (**A**) Deciduous foliage in Buxa Tiger Reserve, West Bengal, India, (**B**) Conifer foliage in Shimla, India, (**C**) Wheat crop in an agricultural area of Himachal Pradesh, India, (**D**) A popsicle in Haridwar, India, (**E**) A candy bar in Shimla, India, (**F**) A tea bun in Buxa Tiger Reserve, West Bengal, India.

Photos taken by Suresh Roy (**A**, **F**), Shaurabh Anand (**C**), Rishabh Bharadwaj (**D**), and Stefano Kaburu (**B**, **E**).

*Radhakrishna, 2016*; *Tang et al., 2016*). They will also consume larval and adult insects, as well as spiders, fish, crabs, birds' eggs, and honeycomb (*Lindburg, 1971*; *Fooden, 2000*). Geophagy (the ingestion of soil or clay) is common in some populations, potentially to prevent gastrointestinal disorders and as a mineral nutritional supplement (*Fooden, 2000*; *Mahaney et al., 1995*).

Generally, rhesus macaques prefer foods that are easily obtained and consumed, but will shift their diet to less appealing or accessible food items (i.e. 'fallback foods') readily when necessary (*Sengupta and Radhakrishna, 2015*; *Tang et al., 2016*). However, how this variation in diet and habitat influences rhesus macaque life-history, health, and fitness across their natural range is poorly understood (*Box 2 Q1*).

In very recent times, rhesus macaques have evolved alongside an intense and rapid form of environmental disturbance associated with human agriculture and urbanization. While these same processes have caused the ranges of many other primate species to progressively shrink, rhesus macaques thrive on agricultural and urban land (*Richard et al., 1989*). In agricultural areas, rhesus macaques have shifted their diet to cultivated crops (*Anand et al., 2018*; *Pirta et al., 1997*; *Rao et al., 2002*), and in more urban areas a considerable portion of rhesus macaque diet is either stolen from humans or is intentionally provisioned by people who offer food for religious or cultural reasons (*Sengupta and Radhakrishna, 2020*; *Sarker et al., 2008*; *Teas et al., 1980*). They likely have developed a preference for anthropogenic food as these are calorie-rich, easily digestible, and abundantly and predictably available (*Sengupta and Radhakrishna, 2018*). The urban success of the species provides an excellent opportunity for studies of the traits that allow for successful coevolution of a species with human societies (*Box 2 Q2*), as well as the impact of urban environments and contemporary diets on physiology, health and ecological functions.

Despite this urban and Anthropocene success, the species is not free from threats such as habitat degradation, poaching and indiscriminate use in biomedical research and commercial trade (*Radhakrishna and Sinha, 2011*). Being categorized as 'Least Concern' on the IUCN Red List effectively excludes rhesus macaques from many conservation plans. Hence wild populations of rhesus macaques should be monitored closely to

## Box 2. Outstanding questions about the natural history of the rhesus macaque.

Despite rhesus macaques' popular use in biomedical research and the relatively thorough study of their behavior and social organization, there are many questions remaining about the natural history of the species, especially in their natural range. Below we described some of these pertinent questions, and discuss their importance relative to rhesus macaque ecology and to evolutionary ecology more broadly.

Q1. How do differences in ecology and diet influence rhesus macaque life history? Rhesus macaques are able to survive over a considerable range of habitats and diets (see "Ecology"). What is less clear is how variation in habitat and diet influences rhesus macaque life-history, reproduction, social behavior, and lifespan. Given the considerable range of habitats over which rhesus macaques are found, they offer great promise for understanding the links between ecology and intra-specific trait variation.

Q2. Does the success of urbanized rhesus macaques represent behavioral plasticity or rapid evolutionary adaptation? The behavior of rhesus macaques in more urbanized areas can be considerably different from non-urban populations in terms of their activity budgets, sociality, social structure, and temperaments (*Seth and Seth, 1985*; *Jaman and Huffman, 2013*; *Kamrul et al., 2013*; *Kaburu et al., 2019*; *Capitanio and Mason, 2019*). In a series of classic studies on rhesus macaques captured from forest and urban areas and compared for several behavioral phenotypes, it was observed that the urban monkeys were more active, manipulated objects more, were more responsive to stimuli of higher complexity values, and were more aggressive, though not better in their problem solving capabilities, than the forest monkeys (*Singh, 1968b*; *Singh, 1968a*; *Singh, 1969*). There is a great deal of interest in both the fields of evolutionary ecology and applied conservation science in understanding the relative roles of behavioral plasticity versus evolutionary change in allowing a species to achieve success in human-modified environments (*Atwell et al., 2012*; *Miranda et al., 2013*). However, our understanding of these processes in rhesus macaques, and even non-human primate species more broadly, is not yet well established (*Miranda et al., 2013*). The large distribution of rhesus macaques over both highly urban habitat and habitat untouched by human development provides an exceptional opportunity to investigate questions central to the roles of evolution and plasticity in determining behavior in urbanized landscapes. This would require pedigree relationships for populations from both urban and non-urban areas, and using quantitative genetic modeling to assess if trait(s) associated with urbanization have shifted due to underlying genetic changes or can be attributed only to environmental (non-genetic) shifts in phenotype.

Q3. Since sociality is apparently so widely beneficial for rhesus macaque fitness, why does variation in sociality persist? A marked characteristic of rhesus macaques is their high levels of sociality, and there are many measurable benefits to sociality for individual fitness (see "Social Style and Dominance"). Despite this, considerable inter-individual variation in the frequency of social interaction and resulting strength of social connectedness persists in rhesus macaque populations (*Brent et al., 2013*; *Brent et al., 2017b*; *Madlon-Kay et al., 2017*). At the extreme end of the spectrum there are remarkably asocial phenotypes; solitary males, living independently of nearby groups have been observed across varied habitats, environmental conditions, and population densities in the rhesus macaque natural range (*Fooden, 2000*). This low sociality phenotype is likely not simply a consequence of environmental conditions, since low-sociality behavioral traits have been shown to be heritable in rhesus macaques (*Brent et al., 2014*; *Brent et al., 2017b*), suggesting that asociality is perhaps adaptive under certain conditions. Interestingly, recent sequencing of the rhesus macaque genome has shown that the species expresses genetic variants which have human analogues that are associated with neurodivergent sociality phenotypes such as autism spectrum disorder (*Warren et al., 2020*). More investigation of selective forces acting on asocial behavior and other apparently

neurodivergent social behavior is needed to better understand the phenomenon of asociality within rhesus macaque societies. These selective forces are likely complex in that they may be non-linear, multi-trait dependent, frequency-dependent, and involve gene-by-environment interactions. In addition to selective forces on sociality, selective forces on consistent individual differences in behavior more broadly (i.e. 'personality') also warrants further investigation, particularly in the context of rhesus macaques' successful infiltration into urban habitats. There is interest from both a conservation biology perspective, and a fundamental evolutionary ecology perspective, to understanding the evolutionary causes and ecological consequences of animal personality in species' success in urbanized environments (*Miranda et al., 2013*; *Lapiedra et al., 2017*). As a highly successful urban niche colonizer, rhesus macaques would make an excellent study species to investigate this phenomenon. While rhesus macaque personalities appear to be heritable (*Brent et al., 2014*), further research is needed to investigate if rhesus macaques' success in urban environments is associated with plastic and/or evolutionary changes in personality.

ensure that they do not have the same fate as other so-called 'common' macaque species such as the bonnet macaque (*Macaca radiata*) and the longtailed macaque (*Macaca fascicularis*), both of which have suffered major declines in their population size over the past few decades (*Eudey, 2008*; *Erinjery et al., 2017*).

## Social organization

Rhesus macaques live in multi-male multi-female groups with a polygynandrous mating system. Groups have a female-biased adult sex ratio; on average there is one mature male per three mature females per group (*Fooden, 2000*). Male rhesus macaques disperse from their natal group to join a neighboring group around the time they reach sexual maturity, but females typically remain in their natal group for their entire lives, such that social groups are stably composed of matrilines (i.e. families of females related through the maternal line; *Neville, 1968*; *Lindburg, 1971*; *Sade, 1972*; *Teas et al., 1980*; *Maestripieri and Hoffman, 2012*). Male group membership is not fixed through adulthood, as males will periodically leave one group to join another, and sometimes males will not be affiliated with any group at all (see *Box 2 Q3*; *Lindburg, 1969*; *Boelkins and Wilson, 1972*; *Drickamer and Vessey, 1973*).

*Fooden, 2000* synthesized studies from nine countries across a range of habitat types which included 1188 rhesus macaque groups and found that for non-provisioned and minimally-provisioned wild rhesus macaques the average group size was 32, and the maximum recorded was 250. Food provisioning of wild populations results in a substantial increase in group size,

with the average being 77 and the maximum recorded being 1045 individuals (*Fooden, 2000*). Home ranges between adjacent groups typically overlap, and group fissions will occasionally happen and typically occur along matrilines, with female family members almost always staying together (*Lindburg, 1971*; *Widdig et al., 2006*; *Sueur et al., 2010*).

## Social style and dominance

Rhesus macaques are notable for their high frequency and severity of aggression and absence of reconciliation between conspecifics when compared to other macaque species, which leads to a social organization that can be described as both highly "despotic and nepotistic" (*Thierry, 2007*).

Steep linear dominance hierarchies are formed within a social group, for both males and females separately (*Datta, 1988*; *Missakian, 1972*). For females, dominance rank is relatively stable and is determined first by matrilineal group, and second by age, with younger females outranking their older sisters (*Berman, 1986*; *Missakian, 1972*). When males immigrate to a new social group they generally enter at the bottom of the social hierarchy and rise through queuing, such that group tenure-length is the primary determinant of male rank (*Manson, 1998*; *Maestripieri and Hoffman, 2012*; *Higham and Maestripieri, 2014*). This is in striking contrast to many other species of macaque, in which there is top-entry contest competition over dominance where males directly compete for the highest rank within a social group (e.g. crested macaques, *Marty et al., 2016*; lion-tailed macaques, *Kumar et al., 2001*). There have been occasional observations

of male rhesus macaques rising in dominance rank through coalitionary aggression, but these are thought to be rare (*Higham and Maestripieri, 2010*).

These steep dominance hierarchies are common in males in most macaque species, but are typically less pronounced among females of many other macaque species (*Thierry, 2007*). The prevalence of steep linear female dominance hierarchies among female rhesus macaques is posited to be related to a number of other components of social organization within the species, including a high bias in forming social bonds predominantly with kin, influence on maternal care style, and higher degree of modularity in social interactions (*Sueur et al., 2011*; *Thierry, 2007*; *Thierry et al., 2008*). Antagonistic dyadic interactions are common both within and between the sexes, and generally result in a 'winner' who asserts dominance through an aggressive behavior (e.g. lunging, biting, or chasing), and a 'loser' who responds with a submissive behavior (e.g. cringing, fear-grimacing, or fleeing; *Blomquist et al., 2011*; *Missakian, 1972*; *Sade, 1967*). Once the dominance hierarchy is established, repeated antagonistic interactions typically function to reinforce, but sometimes challenge, the dominance relationship between two individuals.

In both females and males, dominance rank is an important modulator of individual life-history and fitness. In females, higher dominance rank has been associated with greater annual fertility, survival, earlier maturation of up to half a year, and shorter inter-birth interval (*Drickamer, 1974*; *Bercovitch and Berard, 1993*; *Blomquist, 2009*; *Blomquist et al., 2011*). In males, dominance rank is positively correlated with mating and reproductive success, but this can be highly variable by population and year (*Manson, 1992*; *Bercovitch and Nürnberg, 1997*; *Berard, 1999*; *Widdig et al., 2004*; *Dubuc et al., 2011*; *Dubuc et al., 2014b*). In addition to the competitive processes that shape rhesus macaque society, affiliative social behavior in rhesus macaques is also extremely common and a fundamental component of their social style. Social grooming, or allogrooming, where one individual uses their fingers to remove debris and skin parasites from another is foundational to strong social bonds between conspecifics in rhesus macaques. Individuals that spend a relatively large amount of time in close proximity, in physical contact, and grooming one another are typically characterized as having strong social bonds (*Maestripieri and Hoffman, 2012*).

Affiliative social relationships have been shown to have considerable influence on individual life-history, health, and fitness. Various measures of social stability, integration, and connectedness have been shown to affect a wide variety of physiological metrics in rhesus macaques, including the sympathetic nervous system, white blood cell count and fecal glucocorticoid concentrations (*Capitanio and Cole, 2015*; *Pavez-Fox et al., 2021*; *Brent et al., 2011*). Strong social bonds may also provide indirect benefits such as receiving coalitionary social support when antagonized by another group member (*Kulik et al., 2012*; but see *O'Hearn et al., 2022*). As in a number of other primate species (e.g. *Silk, 2007*), measures of increased social affiliation are positively related to both reproductive output and survival probability (*Brent et al., 2013*; *Ellis et al., 2019*; *Brent et al., 2017a*).

Early-life social environment, and particularly early-life social adversity, may be particularly important in predicting adulthood sociality, health and fitness. Rhesus macaque early-life adversity is predictive of both sociality variables in adulthood, such as rates of aggression, as well as physiological metrics that may be associated with health and fitness including hypothalamic-pituitary-adrenal axis dysregulation (*Barr et al., 2003*; *Dettmer et al., 2017*) and DNA methylation profile (*Massart et al., 2016*). Sociality affects diverse physiological systems, and ultimately health, life-history, and fitness in rhesus macaques in significant and long-standing ways (*Chiou et al., 2020*; *Snyder-Mackler et al., 2016*). The varied and extensive influence of sociality on rhesus biology provides an excellent system for understanding the underlying evolutionary ecology of the sociality-health-fitness axis (*Capitanio and Cole, 2015*; *Cavigelli and Caruso, 2015*; *Hawkley and Capitanio, 2015*; *Kappeler et al., 2015*; *Nunn et al., 2015*).

## Reproduction

Reproduction, which encompasses both the production and nourishing of offspring, is limited, and thus shaped, by energy constraints. How energy is allocated in female reproductive effort can be characterized on a spectrum from 'capital breeders', where adequate available capital (i.e. energy stores) trigger and fuel reproduction, to 'income breeders', where reproductive success is contingent on the incoming flow of resource acquisition at the time of reproduction, rather than stored reserves (*Brockman, 2005*; *Jönsson and Jonsson, 1997*). Rhesus macaques

are considered 'relaxed income breeders' in that they display primarily the characteristics of 'income breeders', but they are not 'strict' in the sense that endogenous conditions such as body mass do play some role in determining reproductive success (*Brockman, 2005*; *Tian et al., 2013*). This characterization provides a basis for understanding many characteristics of their reproductive behavior.

As income breeders, rhesus macaques time birth and lactation with the months of the year where food availability is highest, as they require the highest energetic intake during this time. Consequently, mating occurs approximately in the fall and winter, and the birth season occurs over the spring and summer (*Fooden, 2000*). Interestingly, a group of rhesus macaques which were breeding seasonally at Cayo Santiago lost seasonality when brought to German Primate Center at Gottingen and housed in controlled temperature and photoperiods. However, in the subsequent years when the group was released into a large outdoor enclosure, seasonality reappeared (*Kaumanns et al., 2013*).

The combination of large group sizes and reproductive seasonality leads to females within a population displaying synchrony in their fertility cycles and is thought to be the driver of weak direct male-male competition, such as through physical combat, and strong indirect male-male competition, such as through female mate choice (*Gogarten and Koenig, 2013*; *Kutsukake and Nunn, 2009*; *Ostner et al., 2008*). High female fertile phase synchrony lowers the ability for males to monopolize females, creating a relatively weak competitive environment which leads to relatively low male reproductive skew where reproductive success is not monopolized by only one or a small number of males, and selects for limited body and canine size dimorphism and large relative testis volume (*Dubuc et al., 2014b*; *Harcourt et al., 1981*; *Dubuc et al., 2014a*).

Rhesus macaques exhibit strong evidence of direct female mate choice, with females playing an active role in soliciting mating from a preferred partner (*Maestripieri and Hoffman, 2012*; *Bercovitch, 1997*). Females prefer to mate with novel males (i.e. those that have recently immigrated to their social group), middle-aged males, and males with darker red faces (*Maestripieri and Hoffman, 2012*; *Dubuc et al., 2014a*). Since the highest ranking males in a social group typically have a long tenure within the group and are of an older age class, female choice is likely to be an additional driver of the relatively weak reproductive skew in male rhesus macaques,

by increasing the reproductive success of lower ranking males (*Bercovitch, 1997*).

Females will typically enter multiple ovarian cycles during the mating season, each lasting 30 days. Consortships lasting several hours to several days are formed between a fertile female and a male partner; the pair remains in close proximity during which time the pair engages in multi-mount copulations (*Bercovitch, 1997*; *Higham et al., 2011*). Females are likely to have more than one male consort, averaging three to four mating partners within a single fertile period (*Manson, 1992*; *Lindburg, 1971*; *Bercovitch, 1997*). Males with higher dominance rank engage in consorts of longer duration, while lower ranking males will either engage in shorter consorts or 'sneak' matings (*Bercovitch, 1997*; *Higham et al., 2011*).

In provisioned populations (both captive and free-ranging), females typically have their first offspring between the ages of 3 and 5, and produce one offspring per year until reproduction slows as a result of senescence, typically around age 17 (*Lee et al., 2021*; *Pittet et al., 2017*; *Wilson et al., 1983*). The age-specific demography of reproduction is not well-studied in wild, non-provisioned populations (see *Fooden, 2000*), however in other macaque species provisioning accelerates development, and the timing and pace of female reproduction (; *Fa, 1984*). One study of wild negligibly-provisioned Chinese rhesus macaques found the average age of first birth to be 4.9 years old, approximately a year later than the average seen in food-supplemented populations (*Tian et al., 2013*).

Gestation length in rhesus macaques is approximately 166 days, and sex ratio at birth is not significantly different from 1:1 (*Fooden, 2000*; *Tian et al., 2015*). Annual birth rate (births/sexually mature female) varies largely by population, and in wild-feeding populations annual birth rate per female is between 0.43 and 0.91 (*Fooden, 2000*), with lower birth rates associated with populations at the Northernmost edge of the natural range (e.g. *Wenyuan et al., 1993*).

Environmental perturbation has also been shown to influence female rhesus macaque fertility. For example, on the island population of Cayo Santiago, female reproductive output dropped significantly in years when the island was hit by a hurricane (*Morcillo et al., 2020*). Additionally, matrilineal overthrows, a rare event in rhesus macaque societies where the highest-ranking matriline is violently overthrown by a lower ranking matriline, has been shown to result in significant infant loss for both the deposed and

the attacking matrilines (*Ehardt and Bernstein, 1986*; *Dettmer et al., 2015*).

## Mortality

The primary sources of mortality for rhesus macaques are unclear, but are likely to vary by population. For one population at the northernmost limit of the species range, the main source of mortality was starvation or winter exposure to cold temperatures (*Wenyuan et al., 1993*), but this is not reported in warmer latitudes. Mortality from predation also varies by habitat, with juveniles being the main prey. Recorded predators of rhesus macaques include snakes (*Seth and Seth, 1983*), hawks and eagles (*Lindburg, 1971*; *Wenyuan et al., 1993*), dogs (*Lindburg, 1971*), and large cats such as tigers and leopards (*Lovari et al., 2015*). Physical fighting between rhesus macaques can also be deadly, and mortality due to wound complications such as tetanus infection following physical altercations may be a common cause of death (*Rawlins and Kessler, 1982*).

Annual mortality has been reported between 2% and 10% for adult rhesus macaques, and between 7% and 32% for infants across different wild populations (*Fooden, 2000*). Males have the highest mortality rate during the mating season, and females have the highest mortality rate during the birth season, reflecting the sex-specific costs of reproduction (*Hoffman et al., 2008*).

Due to the logistical difficulties inherent in measuring lifespan in long-lived animals in the wild, average and maximum lifespans in wild rhesus macaques are not known with confidence. In captive populations under controlled conditions, rhesus macaques typically live for 25–30 years, and maximum lifespan has been recorded as 40 years (*Colman et al., 2009*). Under the less controlled conditions of the free-ranging population on Cayo Santiago where rhesus macaques are provisioned and predator-free, but not given medical care, most adult females die between 15 and 25 years of age, with the maximum recorded lifespan being 31 (*Hoffman et al., 2010*; *Maestripieri and Hoffman, 2011*; *Chiou et al., 2020*). Lifespan in the wild is likely to be considerably shorter than under these conditions and may be comparable to the averages of other cercopithecine monkeys of 10–15 years under natural conditions (*Maestripieri and Hoffman, 2012*).

## Conclusion

Rhesus macaques have not only persisted, but flourished, in the Anthropocene. They occupy the largest geographic range of any non-human primate species and display a remarkable level of ecological flexibility. Their persistence across a broad environmental range has long been recognized as a defining feature of the species, as evidenced by their description as 'weed macaques' (*Maestripieri, 2008*; *Richard et al., 1989*). While urbanization often results in the pushing out of endemic wildlife (*McKinney, 2008*; *Savard et al., 2000*), the opposite is true of rhesus macaques, as the species has been known to prefer habitats in urban and agricultural areas, leading to them increasingly being considered a 'pest' species (*Saraswat et al., 2015*; *Sharma and Acharya, 2017*).

The rhesus macaque remains one of the most intensively studied non-human primates, but fundamental questions about the species remain (*Box 2*). Given the frequency of use of rhesus macaques as model organisms in biomedical and behavioral research, a greater understanding of rhesus macaque biology in the wild is needed to better understand and contextualize laboratory research. Additionally, much of the current research on the behavior of rhesus macaques under free-ranging settings has occurred on Cayo Santiago, a population free of natural predators, and where monkeys are provisioned (*Rawlins and Kessler, 1986*; *Box 1*). Given the vast variation in habitats and diets of rhesus macaques, combined with the considerable ranges of group sizes and large genomic variation, there is likely to be undocumented intra-specific variation in behavioral and other phenotypes. This diversity makes the rhesus macaque an excellent model system for investigating multiple components of ecology and evolution, which has been similarly observed for another Papionin group, the baboons, in a previous article in this same series (*Fischer et al., 2019*). New initiatives are currently underway to genotype thousands of rhesus macaques, which will open the door to studying novel genotype-phenotype relationships in the premier non-human primate model for human health and disease.

## Acknowledgements

We sincerely thank all the photographers who provided photographs this manuscript, including Suresh Roy, Rishabh Bharadwaj, Stefano Kaburu and Shaurabh Anand. We thank Helena Pérez Valle, Amanda Dettmer, Yafei Mao, and one

anonymous reviewer, who provided insightful comments on a previous version of this manuscript.

**Eve B Cooper** is in the Department of Anthropology, New York University, New York, United States
eve.cooper@nyu.edu
http://orcid.org/0000-0003-3804-6285

**Lauren JN Brent** is in the Department of Psychology, University of Exeter, Exeter, United Kingdom
http://orcid.org/0000-0002-1202-1939

**Noah Snyder-Mackler** is in the School of Life Sciences, Center for Evolution and Medicine, School of Human Evolution and Social Change, and the ASU-Banner Neurodegenerative Disease Research Center, Arizona State University, Tempe, United States
http://orcid.org/0000-0003-3026-6160

**Mewa Singh** is in the Biopsychology Laboratory, University of Mysore, Mysuru, India
http://orcid.org/0000-0002-9198-0192

**Asmita Sengupta** is at the Ashoka Trust for Research in Ecology and the Environment, Bengaluru, India, and the National Institute for Advanced Studies, Bengaluru, India
http://orcid.org/0000-0002-2477-7290

**Sunil Khatiwada** is in the Institute of Genetics and Animal Biotechnology, Polish Academy of Sciences, Magdalenka, Poland
http://orcid.org/0000-0003-1807-7375

**Suchinda Malaivijitnond** is in the Department of Biology, Chulalongkorn University, Bangkok, Thailand and in the National Primate Research Center of Thailand, Chulalongkorn University, Saraburi, Thailand
http://orcid.org/0000-0003-0897-2632

**Zhou Qi Hai** is in the Key Laboratory of Ecology and Rare and Endangered Species and Environmental Protection, Guangxi Normal University, Guilin, China
http://orcid.org/0000-0002-2832-5005

**James P Higham** is in the Department of Anthropology, New York University, New York, United States
http://orcid.org/0000-0002-1133-2030

*Author contributions:* Eve B Cooper, Conceptualization, Writing – original draft; Lauren JN Brent, Funding acquisition, Writing – review and editing; Noah Snyder-Mackler, Funding acquisition, Writing – review and editing; Mewa Singh, Writing – review and editing; Asmita Sengupta, Writing – review and editing; Sunil Khatiwada, Writing – review and editing; Suchinda Malaivijitnond, Writing – review and editing; Zhou Qi Hai, Writing – review and editing; James P Higham, Supervision, Funding acquisition, Writing – review and editing

*Competing interests:* The authors declare that no competing interests exist.

## Funding

| Funder | Grant reference number | Author |
| --- | --- | --- |
| National Institutes of Health | R01-AG060931 | Eve B Cooper Lauren JN Brent Noah Snyder-Mackler James P Higham |

The funders had no role in study design, data collection and interpretation, or the decision to submit the work for publication.

### Decision letter and Author response
Decision letter https://doi.org/10.7554/eLife.78169.sa1
Author response https://doi.org/10.7554/eLife.78169.sa2

## Additional files

### Supplementary files
• Transparent reporting form

### Data availability
No new data was generated for this article.

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
