## [Decision Letter]

**Decision letter after peer review:**

Thank you for submitting your article "The Natural history of Model Organisms: The rhesus macaque as a success story of the Anthropocene" to *eLife* for consideration as a Feature Article. Your article has been reviewed by three peer reviewers, and the evaluation has been overseen by myself, Helena Pérez Valle, as a member of the *eLife* Features Team. The following individuals involved in review of your submission have agreed to reveal their identity: Amanda Dettmer and Yafei Mao.

The reviewers and editors have discussed the reviews and we have drafted this decision letter to help you prepare a revised submission.

The *eLife* Features Editor will also contact you separately about some editorial issues that you will need to address.

Summary:

This article is a thorough and accessible review of the natural history of the rhesus macaques. In particular, the review provides important updates for the 2020s, mainly focusing on the macaques' evolutionary history, diversity, ecology, and social organization, but also describing research into the genetics and epigenetics of the species. The authors do a commendable job at presenting multiple components of rhesus macaque life history in a digestible, engaging and logical way, thus making a strong argument for this species' success during the Anthropocene.

Essential revisions:

1. Please include a figure showing a time calibrate tree diagram (phylogeny) in the "Evolutionary History" section of the manuscript to better describe the divergence of different macaques and their split time.

2. Please rework the section "Evolutionary History" so that, rather than listing divergence dates and different species or higher taxa (e.g. Papionini), it serves other parts of the article. Please emphasize the following:

a) how rhesus macaques are part of a large, geologically recent radiation of macaques.

b) how rhesus macaques are an unusually successful species even compared to their closest relatives within the genus.

c) how rhesus macaques have a history of hybridization that anthropologists might use to consider hominin hybridization/ancient admixture.

3. In the section on "Reproduction", starting at line 396, please expand your discussion about the effects of environmental perturbation on female rhesus fertility by also describing the influences of troop conflict (specifically, severe aggression/fighting and overthrows) on female fertility/reproduction and group fitness. For this discussion, the authors can look towards the following references (some are in other species but all are in cercopithecine primates):

Chance MRA, Emory GR, Payne RG. 1977. Status referents in long-tailed macaques [Macaca fascicularis]: precursors and effects of a female rebellion. Primates 3:611-632.

Dettmer, A. M., Woodward, R. A., and Suomi, S. J. 2015. Reproductive consequences of a matrilineal overthrow in rhesus monkeys. American Journal of Primatology, 77(3), 346-352.

Ehardt CL, Bernstein IS. 1986. Matrilineal overthrows in rhesus monkey groups. International Journal of Primatology 2:157-181.

Nash LT. 1974. Parturition in a feral baboon [Papio anubis]. Primates 2-3:279-285.

Ruiz-Lamibdes AV, Aure B, Caraballo G, Platt ML, Brent LJ. 2013. Matrilineal overthrow followed by high mortality levels in free‐ranging rhesus macaques. American Journal of Primatology 75:98-99.

Wilson ME, Gordon TP, Bernstein IS. 1977. Timing of births and reproductive success in rhesus monkey social groups. Journal of Medical Primatology 4:202-212.

4. In figure 1, please show the geographic distributions of other macaque species to highlight the wide range of rhesus macaques' habitats. If possible, also include apes' geographic distribution as a comparison.

5. Please include a figure plotting a world map with the primate centers that keep rhesus macaques, to show the impact of this species in biomedical research.

6. If possible, please expand the discussion regarding rhesus macaques' comparatively large natural range by providing information regarding the population size or effective population size (change in population size) of rhesus macaques compared to other non-human primates during the Anthropocene.

7. In reference to lines 93-96, lines 150-154, and lines 170-176, please provide a quantitative description of the genetic differences between Chinese macaques, Indian macaques and Cayo macaques; for example, please mention what the Fst value between the different groups is.

8. Please include a discussion of whether genetic evidence supports the population bottleneck of Cayo macaques, based on the Cayo macaque history.

9. Please provide specific examples of genotype-phenotype relationships in rhesus macaques when they are mentioned them in the article.

---

## [Author Response]

Essential revisions:1. Please include a figure showing a time calibrate tree diagram (phylogeny) in the "Evolutionary History" section of the manuscript to better describe the divergence of different macaques and their split time.

We have added a phylogeny of the 24 species currently recognized under the *Macaca* genus as Figure 1 of the paper. This phylogeny was constructed using TimeTree, a public knowledge-base for information on time-calibrated phylogenies (timetree.org; Kumar et al., 2017). The phylogeny produced using Timetree accurately reflects the most recent peer-reviewed phylogenies constructed for rhesus macaques (e.g. Osada et al., 2021), while also including the complete genus.

2. Please rework the section "Evolutionary History" so that, rather than listing divergence dates and different species or higher taxa (e.g. Papionini), it serves other parts of the article. Please emphasize the following:a) how rhesus macaques are part of a large, geologically recent radiation of macaques.b) how rhesus macaques are an unusually successful species even compared to their closest relatives within the genus.c) how rhesus macaques have a history of hybridization that anthropologists might use to consider hominin hybridization/ancient admixture.

We have substantially reworked this section in order to highlight all three of these points (Lines 130 – 202). We have removed specifics about divergence dates for different species. We address each point by: (a) discussing the large, geologically recent radiation of macaques, and by naming the most closely related macaque species, for which we now provide a phylogeny as Figure 1; (b) highlighting studies which support the claim that rhesus macaque effective population size is larger than other closely related macaque species (Line 183 – 188) and; (c) explicitly stating this point at the end of the third paragraph of this section (Line 173).

3. In the section on "Reproduction", starting at line 396, please expand your discussion about the effects of environmental perturbation on female rhesus fertility by also describing the influences of troop conflict (specifically, severe aggression/fighting and overthrows) on female fertility/reproduction and group fitness. For this discussion, the authors can look towards the following references (some are in other species but all are in cercopithecine primates):Chance MRA, Emory GR, Payne RG. 1977. Status referents in long-tailed macaques [Macaca fascicularis]: precursors and effects of a female rebellion. Primates 3:611-632.Dettmer, A. M., Woodward, R. A., and Suomi, S. J. 2015. Reproductive consequences of a matrilineal overthrow in rhesus monkeys. American Journal of Primatology, 77(3), 346-352.Ehardt CL, Bernstein IS. 1986. Matrilineal overthrows in rhesus monkey groups. International Journal of Primatology 2:157-181.Nash LT. 1974. Parturition in a feral baboon [Papio anubis]. Primates 2-3:279-285.Ruiz-Lamibdes AV, Aure B, Caraballo G, Platt ML, Brent LJ. 2013. Matrilineal overthrow followed by high mortality levels in free‐ranging rhesus macaques. American Journal of Primatology 75:98-99.Wilson ME, Gordon TP, Bernstein IS. 1977. Timing of births and reproductive success in rhesus monkey social groups. Journal of Medical Primatology 4:202-212.

We appreciate the reviewer(s) for providing these references and suggestions. We have added a description of matrilineal take-overs, and specifically cited Ehardt and Bernstein (1986) and Dettmer et al., (2014) as examples of how these events can influence female reproduction, starting at line 484 in the revised version.

4. In figure 1, please show the geographic distributions of other macaque species to highlight the wide range of rhesus macaques' habitats. If possible, also include apes' geographic distribution as a comparison.

We have redone this figure (now Figure 2 in the revised manuscript) using range estimates provided by ICUN Red List for rhesus macaques and 9 of their most closely related sister species (representing a monophyletic group, see Figure 1). One species from this monophyletic group is excluded from the species range map, Macaca leucogenys, a species only discovered in 2015 for which the ICUN Red List does not yet have available range estimates. Since there are 24 macaque species, many with overlapping ranges, we felt a figure including all macaques would be difficult to read and interpret, and so we’ve decided to include only these 9 species most closely related to rhesus macaques here.

5. Please include a figure plotting a world map with the primate centers that keep rhesus macaques, to show the impact of this species in biomedical research.

Unfortunately we do not feel that we are able to do this. There are hundreds of primate centers keeping rhesus macaques globally, across a large number of countries, both within and outside of their natural range. We do not feel that we are able to identify all of these. We are unaware of complete lists of these centers that are publicly available, even on a within-country level. We also note that the locations of captive primate centers are sometimes purposefully hidden from public records for security reasons. (This is the case for some centers holding primates in the US, for example.)

6. If possible, please expand the discussion regarding rhesus macaques' comparatively large natural range by providing information regarding the population size or effective population size (change in population size) of rhesus macaques compared to other non-human primates during the Anthropocene.

We have added details on the effective population size of rhesus macaques starting at line 183. Specifically we write:

“The effective population of rhesus macaques across their entire modern-day range is not known, but current effective population sizes of Chinese and Indian rhesus macaques are estimated at 240,000 and 17,000, respectively (Hernandez et al., 2007).”

We were able to find two relevant studies which compared either effective population size or genetic diversity (interpretable as a proxy for effective population size) between rhesus macaques and a closely related species. Specifically we’ve added, starting at line 185:

“These countries represent only a portion of total rhesus macaques, but even those population sizes alone indicate that rhesus macaques have a substantially larger population than other closely related macaque species in Asia (Marmi et al., 2004; Bonhomme et al., 2009).”

For reasons specific to each of these studies we do not feel it’s appropriate or possible to provide direct numerical comparisons of effective population sizes. Bonhomme et al., (2009) used mitochondrial DNA to estimate effective population sizes of long-tailed and rhesus macaques. Their study found that rhesus have a larger effective population size than long-tailed macaques, however, including the precise numbers reported in the paper would not be appropriate here since effective population size estimated with mitochondrial genome is not going to be comparable to the estimates of effective population size using nuclear genomes (as done in Hernandez et al., 2007, above). Marmi et al., (2004) compared genetic diversity in Japanese and rhesus macaques and determined that the Chinese rhesus macaques have higher genetic diversity, however they do not specifically estimate effective population size.

7. In reference to lines 93-96, lines 150-154, and lines 170-176, please provide a quantitative description of the genetic differences between Chinese macaques, Indian macaques and Cayo macaques; for example, please mention what the Fst value between the different groups is.

We have added Fst estimates for comparisons between Indian and Chinese rhesus macaques starting at line 215. We are unfortunately not aware of any studies quantifying genetic difference (e.g. Fst) between the Cayo Santiago population and wild macaques. However, we have added discussion on the population bottleneck and genetic diversity of the Cayo Santiago population compared with Indian rhesus macaques (see our response below to comment 8, or line 75-83 of the revised manuscript).

8. Please include a discussion of whether genetic evidence supports the population bottleneck of Cayo macaques, based on the Cayo macaque history.

We have added new sentences about the population bottleneck on Cayo Santiago and current genetic diversity. Specifically, we write that:

“The population dipped in the 1950s causing a bottleneck whereby all individuals alive today are the descendants of 15 females which were alive in 1956 (McMillan and Duggleby, 1981). Today the population size is maintained at approximately 1500 individuals. Unsurprisingly, given the relatively small founder population and subsequent population bottleneck, the Cayo Santiago population is less genetically diverse then wild Indian rhesus macaques (Sreetharan et al., 2016). However, despite the reduced genetic diversity, there is no evidence of inbreeding depression in the population with individuals seemingly outbred by disassortative mating (Widdig et al., 2017).” (Line 75-83).

9. Please provide specific examples of genotype-phenotype relationships in rhesus macaques when they are mentioned them in the article.

We have added a brief description of four studies of genotype-phenotype relationships pertaining to both rhesus macaque behavior and physiology, starting at line 241. Specifically, we write that:

“Functional genetic studies on rhesus macaques have been used to identify specific genetic mechanisms underlying a wide variety of physiological and behavioral traits, including the timing of male natal dispersal (Trefilov et al., 2000), the degree of female multimale mating (Trefilov et al., 2005), the oxytocin response and maternal behavior in lactating mothers (Wood et al., 2022), and adaptations to cold climatic conditions at the Northern edge of the species range (Liu et al., 2018). “